# Improving Angular Accuracy of a Scanning Mirror Based on Error Modeling and Correction

**DOI:** 10.3390/s19020367

**Published:** 2019-01-17

**Authors:** Yue Fan, Wenli Ma, Ping Jiang, Jinlong Huang, Kewei Chen, Nian Pan

**Affiliations:** Institute of Optics and Electronics, Chinese Academy of Sciences, 610209 Chengdu, China; mawenli@ioe.ac.cn (W.M.); jiangping@ioe.ac.cn (P.J.); huangjl@ioe.ac.cn (J.H.); awei_bit@163.com (K.C.); wopannian@163.com (N.P.)

**Keywords:** scanning mirror, angular accuracy, eddy current displacement sensor, error model, angle calibration, model correction

## Abstract

Scanning mirrors appear to be key components in optoelectronic systems for line-of-sight (LOS) stabilization. For improving the angular accuracy of a scanning mirror based on the eddy current displacement sensor measurement, an angular error-correction method is proposed and demonstrated. A mathematic angular error model with physical parameters was developed, and the cross-validation method was employed to determine the reasonable order of the Maclaurin series used in the error model, which increased the exactitude and robustness of the correction method. The error parameters were identified by accurately fitting the calibrated angular errors with the error model, which showed excellent error prediction performance. Based on the angular calculation model corrected by the error model, the closed-loop control system was established to obtain accurate deflection angles. Experimental results show that within the deflection angle of ±1.5 deg, the angular accuracy was improved from 0.28 deg to less than 1.1 arcsec, and the standard deviation for six measurements was less than 1.2 arcsec, which indicates that the angle correction method was effective in improving the linearity of the eddy current sensors and reducing the influence of manufacturing and installation errors.

## 1. Introduction

Scanning mirrors have been used as key components in optoelectronic reconnaissance systems, such as InfraRed Search and Track (IRST) and airborne electro-optical (EO) sensors, that provide all-weather, day and night, and high resolution imagery of sea surface, airborne, and ground targets for joint armed forces users [1,2,3]. In such systems, the high-resolution field-of-view (FOV) along with the high coverage rate requires that the line-of-sight (LOS) be stabilized with high accuracy and step-stared at a high rate. A mirror-scanning mechanism is mounted internally in the telescope optical path to meet the fine stabilization and fast step-stare requirements. To compensate image motion, the scanning mirror with one or two rotation axes [4,5] provides a de-scanning action against the continuous scanning of the sensor head including all optical elements and the IR camera. The combination of LOS stabilization and de-scanning reduces the blur and smearing of the scene on the imaging plane during integration, which is particularly important over longer integrations. This improves the resolution of the optical system, thereby increasing image quality and target detection performance [6]. The de-scanning operation allows for efficient panoramic coverage, thus reducing frame-to-frame settling time to fully exploit the high frame rate available on matrix IR cameras, and extending the gimbal hardware life.

Some imaging systems require the LOS jitter to be less than a fraction of a pixel’s instantaneous field of view (IFOV) [7], which is at or below arcsec level. The ability to meet high LOS stabilization accuracy is limited by the ability to measure and correct the angular deflection errors of the scanning mirror in the laboratory environment. The capability of precisely controlling the angular position of the scanning mirror determines the imaging and pointing performance of scanner-based optoelectronic systems. Therefore, measuring and improving the angular accuracy of the scanning mechanism has attracted great attention. 

Paulpandiari et al. presented calibration of the angular motion of a scanning mirror mechanism using a lateral-effect position sensitive photodetector [8]. Korolyov et al. proposed a technique for dynamic error measurement of the angular motion of a scanning mirror using an imaging method [9]. Krishna reported improving the pointing accuracy of a scan mirror using high precision theodolite measurements and an interpolation technique, explaining the experimental phenomenon by mathematical analysis [10]. Though the interpolation curve can fit well with the measured points, there may be large deviations between nearby values and actual values, which will cause large correction errors. Hilkert et al. studied the pointing errors of fast steering mirrors (FSM) by proposing the LOS kinematic and deflection equations, which can be used to simulate the FSM pointing characteristics and formulate control and correction schemes [11]. However, the error correction was not implemented. More recent research has focused on improving the angular accuracy by studying the mathematical relationship between the deflection angle and the characteristics of scanners. Zhang et al. developed a mathematical model to predict the actuation characteristics of the MEMS (Micro-electromechanical Systems) scan mirror and corrected the non-uniformity of the driving force and the nonlinearity between the input voltage and the mirror-plate position based on the model [12]. Li et al. proposed an optimization method to improve the pointing accuracy of a Risley prism scanner by identifying and correcting the physical parameters of the mathematical model based on a genetic algorithm [13].

Various angular measurement methods have been proposed, such as autocollimator [14,15], interferometer [16,17], visual measurement [18,19], and one composed of both the interferometer and autocollimator methods [20]. Among these methods, autocollimators are adopted more often due to their smaller, compact size and lower condition requirements compared with those of the other methods. Autocollimators with wide measurement ranges and high accuracy are used for highly precise measurements of deflection angles [21]. 

In this paper, we propose an angle correction method for improving the angular accuracy of a scanning mirror based on error modeling and correction. A mathematic angular error model with physical parameters was derived from the angular calculation model based on the eddy current displacement sensor measurement, and the order *n* of the Maclaurin series used in the error model was determined by the cross-validation method. Compared to the existing angular error correction methods such as polynomial fitting and interpolation, the proposed method effectively improves the exactitude and robustness of the angle correction, because the error parameter with physical meaning is invariable with measurement conditions and the order *n* is reasonable. In addition, the proposed error model shows excellent performance on error prediction. Experimental results show that the angular accuracy of the scanning mirror is dramatically improved from 0.28 deg to less than 1.1 arcsec by using the corrected calculated angles as the feedback of the position closed-loop control system, indicating that the angle correction method is effective in improving the linearity of the eddy current sensors and reducing the influence of manufacturing and installation errors.

## 2. Angular Error Model

### 2.1. Eddy-Current Sensor-Based Angle-Measurement Principle

As shown in Figure 1, the differential measurement method based on two eddy current displacement sensors is applied to obtain the deflection angle of the scanning mirror. Micro-Epsilon NCDT3010 eddy current sensors with the measurement range of 0.5 mm are used in our system. The two sensors are arranged symmetrically about the rotation axis of the scanning mirror to measure the linear displacements in X direction. Since the mirror plate is made of SiC, which is non-inductive to eddy currents, two induction pads made of ferromagnetic material are stuck to the back surface of the mirror plate, facing the two sensor probes. The output voltages related to the horizontal distances between the sensor probes and the induction pads are transformed to distance data by ADC (Analog-to-Digital Converter). For a counterclockwise deflection of the mirror plate by an angle *θ*, the displacements detected by sensor 1 and sensor 2 are *x*_1_ and *x*_2_, respectively. Then, *θ* is given by
(1)$$\theta =\arctan\left(\frac{x_{2}-x_{1}}{d}\right)$$
where *d* is the distance between the axes of the two sensor probes in the Y direction, which is designed as 12 mm; *x*_1_ and *x*_2_ are given by
(2)$$x_{1}=x_{11}-x_{10}$$
and
(3)$$x_{2}=x_{21}-x_{20}$$
where *x*_10_ and *x*_20_ are the initial distances between the two eddy current sensor probes and the induction pads, respectively, which are both 0.3 mm; *x*_11_ and *x*_21_ are distances between the two eddy current sensor probes and the induction pads when the scanning mirror deflects at angle *θ*, respectively.

Equation (1) is the angular calculation model of the scanning mirror. We can see that the deflection angle of the scanning mirror is related to the distance *d* between the axes of the two sensor probes, but is not related to the distance between the rotational axis of the mirror nor either one of the sensors. So the variations of the center of the two eddy current sensors have no effect on the deflection angles. It is evident that for a translation of the mirror plate or a disturbance to the two sensors in X direction, the same displacement will be detected by the two sensors and be subtracted by *x*_1_ minus *x*_2_. Therefore, by applying the differential measurement method rather than a single sensor measurement, angular errors caused by vibration and thermal deformation in X direction can be avoided.

As shown in Figure 2, the angles calculated by Equation (1) are used as the feedback signals for the position closed-loop control system. By correcting using the control algorithm, the steady-state deviation between the calculated angle and the given angle, which is the ideal angular displacement, reaches zero when the scanning mirror rotates step-by-step. Consequently, the accuracy of the result from the angular calculated model determines the position precision of the scanning mirror.

### 2.2. Error Analysis of the Angular Calculation Model

Since systematic errors and random errors are unavoidable in the calculation factors *x*_1_, *x*_2_, and *d*, the calculated angles will deviate from the actual deflection angles, which will cause the loss of scanning precision. The main task of improving the angular accuracy of the scanning mirror is to identify the systematic errors, those that are invariable or vary regularly over time, and correct the angular calculation model by eliminating the systematic errors.

The systematic errors of the calculation factors *x*_1_ and *x*_2_ mainly originate from the following aspects:
(1)Nonlinearities of the eddy current displacement sensors. The linearity deviation of the Micro-Epsilon NCDT3010 eddy current sensor is less than 1.25 μm, which will cause an angular error of less than 0.01 deg;(2)Nonlinearities caused by the non-vertical relationship between the induction pads and the axes of the sensor probes during scanning. The angular error caused by tilting of the measured object relative to the sensor probe was studied by Tan et al. [22], who gave the nonlinear relationship between the angular error and the deflection angle and compensated the error by establishing the principle error equation.(3)The physical characteristic difference between the ferromagnetic material used for induction pads in our measurement system and that used for adjusting the eddy current sensor controllers in the factory, which will cause extra nonlinearity of the sensor measurements;

For the calculation factor *d*, the systematic error is mainly caused by manufacturing errors of the installation base and installation errors of the sensor probes, which cause the difference between the design value and the actual value. 

Since *θ* is related to *x*_2_ − *x*_1_, take *x*_2_ − *x*_1_ as one factor, and let *x*_2_ − *x*_1_ = *s*. Suppose that Δs and Δd are the systematic errors of *s* and *d*, respectively, the systematic error Δθ of the deflection angle *θ* is given by
(4)$$\begin{array}{ll} \Delta \theta & =\frac{\partial \theta }{\partial s}\Delta s+\frac{\partial \theta }{\partial d}\Delta d\\ & =\frac{d\Delta s-s\Delta d}{s^{2}+d^{2}} \end{array}$$
where *d* is the design value of 12 mm; *s* is the measured value of *x*_2_ − *x*_1_ by the eddy current sensors.

Obviously, Δd is a constant error and will not vary as the measurement conditions change; whereas Δs is varies regularly with variations in s. Though the accurate expression of Δs(s) cannot be obtained, it can be approximated with a Maclaurin series expansion as
(5)$$\Delta s(s)=a_{0}+a_{1}s+a_{2}s^{2}+\ldots +a_{n}s^{n}$$
where
$$a_{0}=\Delta s(0),a_{1}=\Delta s^{\prime }(0),a_{2}=\frac{1}{2!}\Delta s^{\prime \prime }(0),\ldots ,a_{n}=\frac{1}{n!}\Delta s^{\left(n\right)}(0).$$

Substituting Equation (5) into Equation (4), we express Δθ as
(6)$$\begin{array}{ll} \Delta \theta & =\frac{d(a_{0}+a_{1}s+a_{2}s^{2}+\ldots +a_{n}s^{n})-s\Delta d}{s^{2}+d^{2}}\\ & =\frac{d}{s^{2}+d^{2}}\left[a_{0}+(a_{1}-\Delta d/d)s+a_{2}s^{2}+\ldots +a_{n}s^{n}\right] \end{array}$$

Equation (6) is the angular error model that shows the relationship of Δθ and the error parameters Δd, *a*_0_, *a*_1_, *a*_2_… *a_n_*, parameter Δd is the distance error of the two sensor probes, which is invariable with measurement conditions. The priority is to identify the error parameters that uniquely determine the angular error model. If we receive a set of measured values of Δθ, the values of the error parameters can be estimated, which will be discussed in Section 3 and Section 4.

## 3. Calibration of Deflection Angles

Figure 3 shows a schematic diagram of the angular calibration principle. An electrical autocollimator with a measurement range of ±1.5 deg and an accuracy of 1 arcsec is applied to calibrate the deflection angles of the scanning mirror. When the scanning mirror rotates around the Z_1_ axis, the autocollimator detects the angular displacements along the Y_2_ axis. For minimizing the elevation errors, the Y_2_ axis is adjusted to be perpendicular to the Z_1_ axis, with the angular variation detected in the Z_2_ direction not exceeding 10 arcsec when the mirror rotates, reflecting the beam from one edge to the other edge of the FOV of the autocollimator.

By adjusting the elevation and azimuth knobs of the autocollimator, the scanning mirror reflects the collimated beam near the center of the autocollimator FOV when it is at the initial position defined by the calculated angle of 0 deg, and the angle detected by the autocollimator in the Y_2_ direction is recorded as α_0_. Firstly, the scanning mirror rotates to position 1 where the beam is reflected near the edge of the autocollimator FOV. Secondly, the scanning mirror rotates clockwise from position 1 to position 2, which is opposite to position 1 at an interval of 0.1 deg, and simultaneously, the autocollimator detects and records the angular displacements of the scanning mirror one by one. In order to reduce the influence of the random errors on the measurement results, the calibrations are made moving forwards (increment of the angular displacement) and backwards (decrement of the angular displacement) three times, which means each calibration angle point is measured six times, and an average is taken.

The calibrated angle is given by
(7)$$\Theta_{i}={\bar{\alpha }}_{i}-\alpha_{0}$$
where $\Theta_{i}$ is the calibrated angle; *i* is the calculated angle, which is 0 deg, ±0.1 deg, ±0.2 deg, …; $\alpha_{0}$ is the angle detected by the autocollimator when the scanning mirror is at the initial position; ${\bar{\alpha }}_{i}$ is the average value of angles detected by the autocollimator when the scanning mirror rotates at *i*.

Figure 4 shows the lab-built prototype of the angular calibration system. The calibration experiments were carried out in a temperature controlled laboratory (27 ± 1 °C) to minimize the thermal disturbance to the instruments. The electrical autocollimator (TriAngle Large-Field) was fixed to an optical table facing the scanning mirror which was fixed on a lift stage that could be adjusted concentrically with the autocollimator. The scanning mirror was actuated by a pair of voice coil motors, which were parallel-driven by one amplifier synchronously in the opposite direction. The voltage signals from the eddy current displacement sensors (Micro-Epsilon NCDT3010) were processed by the sensor controllers, which were connected by a synchronous cable to enable the synchronicity of the two sensors. During the calibration, the scanning mirror rotated step-by-step according to the deviations between the given angles and the non-corrected calculation angles that were computed in the controller (NI CRIO-9034).

Figure 5 shows the calibration results. The relationship between the calculated angle and the calibrated angle is shown in Figure 5a, in which “Calculated angle” was calculated by using Equation (1), and “Calibrated angle” is the actual deflection angle calibrated by the autocollimator, which is defined by Equation (7). From Figure 2, we can see that the scanning mirror deflects according to the given angles, which are equal to the calculated angles. Ideally, the calculated angles should be equal to the calibrated angles, and the relationship between them can be represented by the “Ideal line” in Figure 5a, which shows a slope of 45 deg. However, due to the systematic errors of the calculation factors *x*_1_, *x*_2_, and *d* analyzed in Section 2.2 and some random errors, the calculated angles deviate from the actual deflection angles, namely, the calibrated angles do not equal the calculated angles, which caused the deviation between the calibrated data line and the ideal line in Figure 5a. The angular error is given by
(8)$$\Delta \theta_{M}=\theta -\Theta$$
where $\Delta \theta_{M}$ is the calibrated angular error; *θ* is the calculated angle defined by Equation (1); Θ is the calibrated angle. 

The angular error as a function of the difference between the calculated angle and the calibrated angle before model correction is shown in Figure 5b, in which “s” is the difference in the displacements measured by the two eddy current displacement sensors, which is defined in Equation (4). It can be seen that the angular error grows dramatically with the increase of the deflection angle, and the maximum absolute value of the angular error reaches 0.28 deg. The accuracy of the mirror scanning was decreased by the large angular error. To make the scanning mirror work in a large range without losing the accuracy of the LOS locating, it is necessary to correct the deflection angles.

From Figure 5b, we can find the consistence between the angular error model proposed by Equation (6) and the calibrated error data. The slope of the angular error along the calibration row indicates that there is a linear system error in the calculated angles, which can be represented by the linear terms of $(a_{1}-\Delta d/d)s$ (*s* is much smaller than *d*, so $d/(s^{2}+d^{2})$ is approximately 1); the curved shape indicates that there are nonlinear system errors, which can be represented by the higher terms of $\left(a_{2}s^{2}+\ldots +a_{n}s^{n}\right)$; the zero position error can be represented by *a*_0_.

## 4. Model Correction

### 4.1. Identification of Error Parameters

The consistence between the angular error model and the calibrated angular error indicates that the model can be used to fit the error data. The order *n* of the Maclaurin series affects the fitting quality of the angular error model, namely, too small an *n* value will cause under-fitting so that the model cannot capture the error data characteristics well; on the contrary, too large an *n* value will cause over-fitting that involves the noise data. In order to obtain a reasonable value of *n*, the cross-validation method is employed. The calibrated error data are divided into the training set and the testing set, which account for 70% and 30% of the error data, respectively, and the data in one set never appears in the other set at the same time. The deviations between the calibrated error data and the theoretical errors calculated by Equation (6) in each set can be computed, which are defined as the training error and the testing error, respectively. The two errors in the RMS (Root Mean Square) form are simply given by
(9)$$J_{\mathrm{training}}=\sqrt{\frac{1}{m_{\mathrm{training}}}\sum_{i=1}^{m_{\mathrm{training}}}{\left(\Delta \theta_{M}^{\left(i\right)}-\Delta \theta^{\left(i\right)}\right)}^{2}}$$
and
(10)$$J_{\mathrm{testing}}=\sqrt{\frac{1}{m_{\mathrm{testing}}}\sum_{i=1}^{m_{\mathrm{testing}}}{\left(\Delta \theta_{M}^{\left(i\right)}-\Delta \theta^{\left(i\right)}\right)}^{2}}$$
where $J_{\mathrm{training}}$ and $J_{\mathrm{testing}}$ are the training error and the testing error, respectively; $\Delta \theta_{M}$ is the calibrated angular error defined by Equation (8); Δθ is the theoretical error defined by Equation (6); *m*_training_ and *m*_testing_ are the number of the training data and the testing data, respectively.

From Equation (6), Equation (9), and Equation (10), it is clear that different *n* will produce different training errors and testing errors. Figure 6 shows the training curve and the testing curve as functions of *n*. We can see that the training error and the testing error are large when *n* is less than 3, which indicates that under-fitting may appear. With increasing values of *n*, the two errors both decrease. But when *n* is larger than 9, the testing error increases and is much larger than the training error when *n* is larger than 11, which means there may be the over-fitting. It is evident that when *n* is between 9 and 11, the training error and the testing error both reach the minimum values, and they are very close, which indicates that the values of *n* are reasonable for the angular error fitting. For simplifying the computing, *n* is determined as 9.

The fitting is processed by applying the Nonlinear Least Squares method, and the objective function is given by
(11)$$\begin{array}{l} J(\Delta d,a_{0},a_{1},a_{2},a_{3},a_{4},a_{5},a_{6},a_{7},a_{8},a_{9})\\ =\sqrt{\frac{\sum_{i=1}^{m}{\left(\Delta \theta_{M}^{\left(i\right)}-\Delta \theta^{\left(i\right)}\right)}^{2}}{m}}\\ =\sqrt{\frac{\sum_{i=1}^{m}{\left\{\Delta \theta_{M}^{\left(i\right)}-\frac{d}{s^{(i)2}+d^{2}}\left[a_{0}+(a_{1}-\Delta d/d)s^{\left(i\right)}+a_{2}s^{\left(i\right)}{}^{2}+\ldots +a_{9}s^{\left(i\right)}{}^{9}\right]\right\}}^{2}}{m}} \end{array}$$
where $\Delta \theta_{M}$ is the calibrated angular error, which is defined by Equation (8); Δd, *a*_0_, *a*_1_, *a*_2_, *a*_3_, *a*_4_, *a*_5_, *a*_6_, *a*_7_, *a*_8_, and *a*_9_ are the error parameters; *d* and *s* are defined in Equation (4); *m* is the number of calibration angle points.

By minimizing the objective function defined by Equation (11) with the Trust Region algorithm, the estimates of the error parameters are obtained as follows:
$$\begin{array}{l} \Delta d=0.073,\\ a_{0}=-6.15\times 10^{-5},a_{1}=0.15,a_{2}=-0.039,\\ a_{3}=0.14,a_{4}=0.18,a_{5}=-0.86,\\ a_{6}=-1.21,a_{7}=5.58,a_{8}=2.88,a_{9}=-12.40 \end{array}$$

Actually, the best n-value and most a-parameters are also dependent on the initial maximum deflection angles. For a smaller initial maximum deflection angle, the best n-value is smaller and the higher-order parameters change greatly, however, the zero-order and first-order parameters are almost invariable compared to the larger angles. The error model with the best n-values and a-parameters identified within small deflection angle ranges cannot predict errors of larger deflection angles accurately. Therefore, for precisely calculating the angular errors of all the initial deflection angles, the best n-value and a-parameters should be identified by the entire angular error data within the calibrated angle range.

With the identified error parameters, the angular error model of the scanning mirror is uniquely determined by
(12)$$\Delta \theta =\frac{12}{s^{2}+12^{2}}\left(\begin{array}{l} -6.15\times 10^{-5}+0.14s-0.039s^{2}+0.14s^{3}+0.18s^{4}\\ -0.86s^{5}-1.21s^{6}+5.58s^{7}+2.88s^{8}-12.4s^{9} \end{array}\right)$$

Figure 7a shows the calibrated error fitted by the angular error model. The fitting error of each calibration point is shown in Figure 7b. It can be seen that the maximum fitting error is less than 0.5 arcsec, which indicates that the angular error model fits the error data well. 

From Equation (12) and the accurate fitting of the calibrated errors by the angular error model, we can see that once the linear displacements of the scanning mirror are measured by the eddy current sensors, the deflection angular error can be evaluated by the error model, which indicates that the angular error model is appropriate for error prediction of the deflection angles.

### 4.2. Correction of Angular Calculation Model

Based on Equation (1) and Equation (12), the angular calculation model can be corrected as
(13)$$\begin{array}{ll} \theta & =\arctan\left(\frac{s}{d}\right)-\Delta \theta \\ & =\arctan\left(\frac{s}{12}\right)-\frac{12}{s^{2}+12^{2}}\left(\begin{array}{l} -6.15\times 10^{-5}+0.14s-0.039s^{2}+0.14s^{3}+0.18s^{4}\\ -0.86s^{5}-1.21s^{6}+5.58s^{7}+2.88s^{8}-12.4s^{9} \end{array}\right) \end{array}$$

As shown in Figure 8, the corrected angular calculation model is used in the position closed-loop control system for calculating the accurate deflection angles of the scanning mirror. As the feedback signal, the calculated result is compared with the given angle, and the steady-state deviation between them will be zero with the use of the control algorithm. So the static accuracy of the deflection angle is finally determined by the residual angular error after the correction of the calculation model. When the static angular error is sufficiently decreased, the dynamic accuracy of the angular motion of the scanning mirror mainly depends on the tracking precision of the control algorithm. In other words, the static and dynamic accuracy of the scanning mirror both will be improved by the model correction.

## 5. Results and Discussion

To verify the effectiveness of the angle correction method, we measured the angular accuracy by using the angular calibration system and method proposed in Section 3 after the model correction. The verification experiments were carried out at the same room temperature as the angular calibration experiments. The angle was measured every 0.2 deg instead of 0.1 deg in order to reduce the test time for the “forward” and “backward” direction by use of the autocollimator, and each direction test was conducted three times to reduce the measurement random errors.

The angular errors between the corrected calculation angle and the angle measured by the autocollimator of the six measurements are shown in Figure 9a. Within the measurement range of ±1.5 deg, the angular errors varied within ±2.5 arcsec and showed no slope, indicating that the linear system errors were eliminated. Although the angular errors were greatly reduced, there were still some residual errors, which were mainly caused by the repeatability errors of the eddy current displacement sensors of 0.05 μm (less than 1.7arcsec) and the fitting errors (less than 0.5 arcsec) analyzed in Section 4.

By averaging the error values over the column of each measurement point, the mean error of the six measurements was obtained. As shown in Figure 9b, the absolute value of the mean error was less than 1.1 arcsec within the measurement range, which we greatly decreased, compared to the mean error of 0.28 deg before model correction, indicating that the angle correction method was effective in reducing the systematic errors generated from the sensor nonlinearity, manufacturing, and installation. 

Computing the deviations of all the angular errors from their mean errors over each column made it possible to obtain the standard deviations, which represent the dispersions of the angular errors around their mean errors over each measurement column. As shown in Figure 10, the maximum standard deviation within the measurement range was less than 1.2 arcsec, which indicates that the limit error of measurement, 3*σ* (on the assumption of a normal distribution), must not exceed ±3.6 arcsec. This value includes the contribution of the autocollimator accuracy of less than 1 arcsec, the control system and sensor noises, the disturbances of the environment, and the wear of the screws for fastening the scanning mirror base.

From the above experimental results, we can see that the static performance of the scanning mirror was significantly improved with the proposed error correction method. Therefore, by using the angles calculated by the corrected model as the feedback, a high dynamic position precision of the scanning mirror can be achieved with the closed-loop control system. Limited to the measurement range of the autocollimator, the deflection angles within ±1.5 deg were calibrated and corrected in this paper. In addition, the proposed angle correction method is also suitable for larger angles. In future work, a wider-range measurement method such as the imaging method [13] can be adopted for measuring the position accuracy of larger angles.

## 6. Conclusions

This paper proposed and verified a model-based angle-correction method for improving the angular deflection accuracy of a scanning mirror. An angular calculation model of the scanning mirror was established using the differential measurement method based on two eddy current displacement sensors. Based on error analysis of the sensors and their distance, an angular error model with the physical parameters was derived and the order *n* of the Maclaurin series was determined by the cross-validation method, which increased the exactitude and robustness of the error correction method. A lab-built angular calibration system was established to perform calibration and verification experiments. The error parameters were identified by accurately fitting the calibrated angular errors with the error model, which shows excellent error prediction performance. The angular calculation model was corrected by the identified error model. Experimental results show that by using the corrected calculated angles as the feedback of the closed-loop control system, the angular accuracy was improved from 0.28 deg to less than 1.1 arcsec, and the standard deviation for six measurements was less than 1.2 arcsec, which indicates that the proposed method is effective at improving the linearity of the eddy current sensors and reducing the influence of manufacturing and installation errors. 

## Figures and Tables

**Figure 1 sensors-19-00367-f001:**
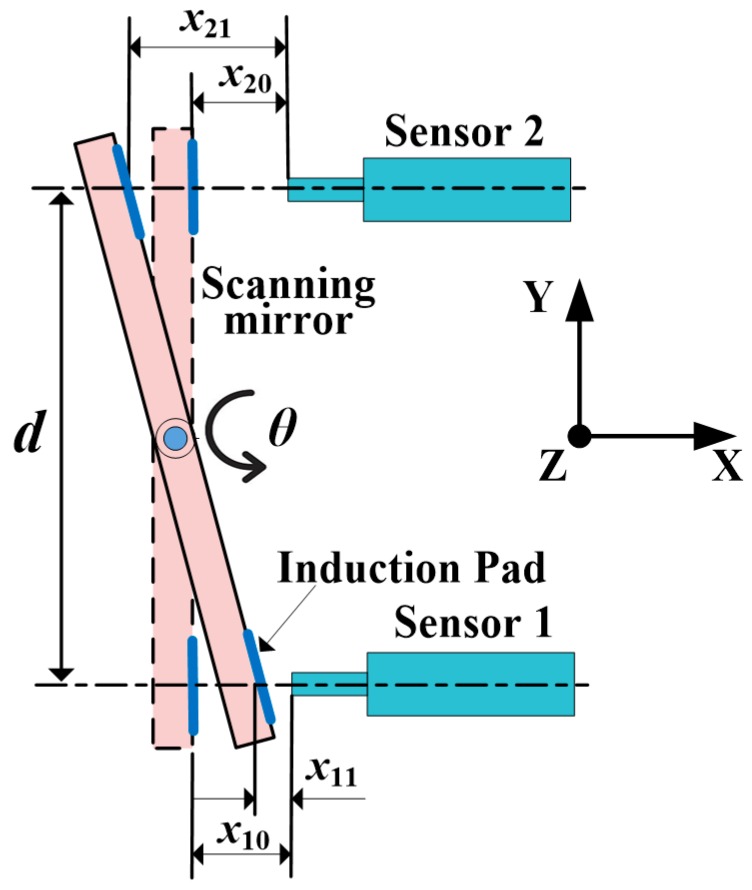
Schematic of the eddy-current sensor-based angle-measurement principle of the scanning mirror.

**Figure 2 sensors-19-00367-f002:**
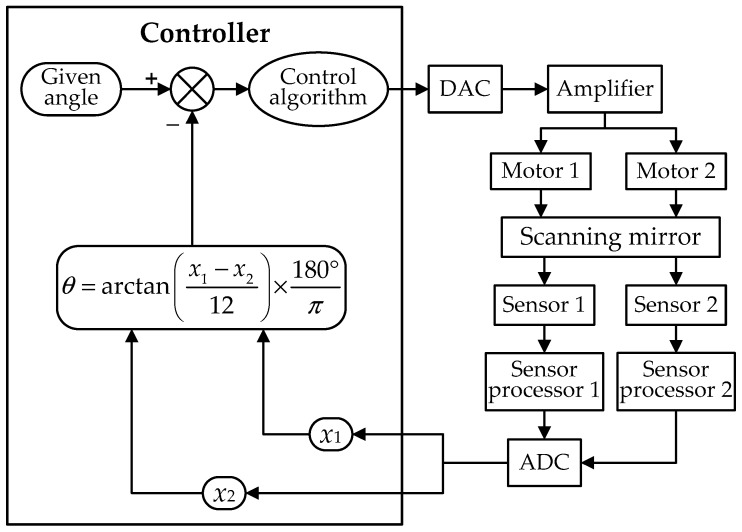
The block diagram of the control system based on the angular calculation model. (DAC is Digital -to-Analog Converter).

**Figure 3 sensors-19-00367-f003:**
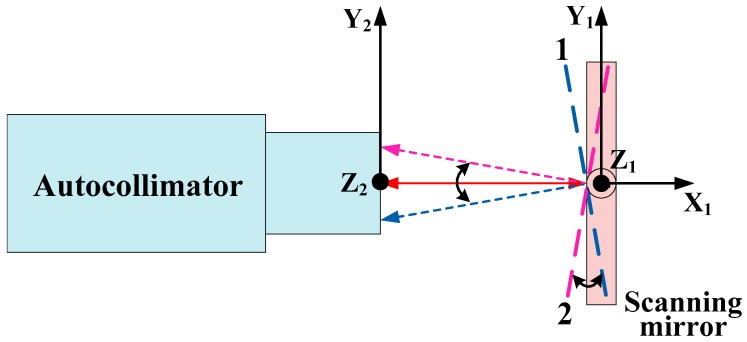
Schematic diagram of the angular calibration principle.

**Figure 4 sensors-19-00367-f004:**
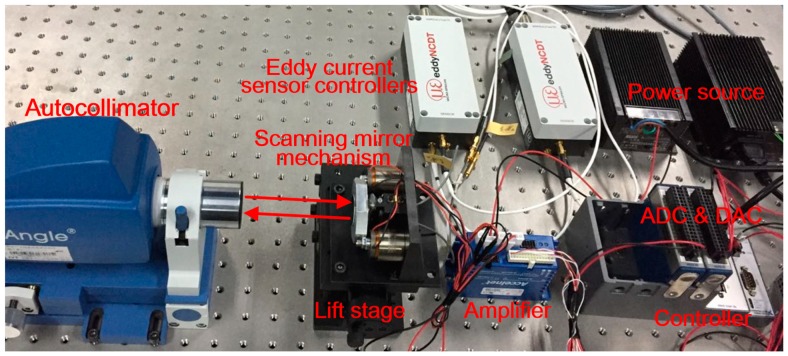
Photograph of the lab-built angular calibration system.

**Figure 5 sensors-19-00367-f005:**
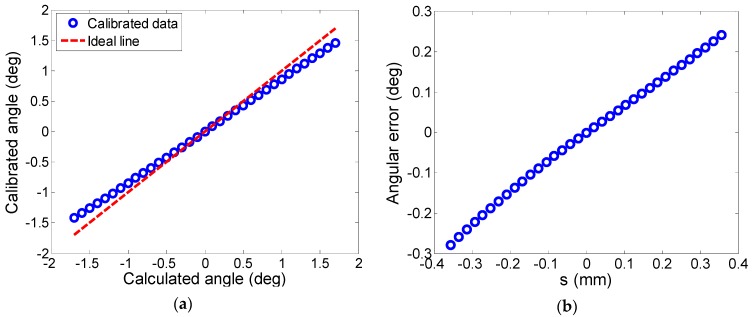
The calibration results: (**a**) The relationship between the calculated angle and the calibrated angle; (**b**) The angular error as a function of the difference between the calculated angle and the calibrated angle before model correction.

**Figure 6 sensors-19-00367-f006:**
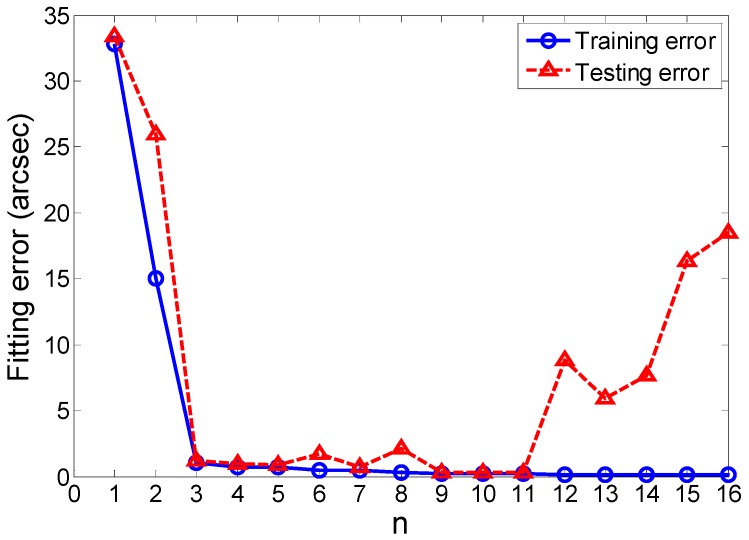
The training errors and testing errors varying with *n*.

**Figure 7 sensors-19-00367-f007:**
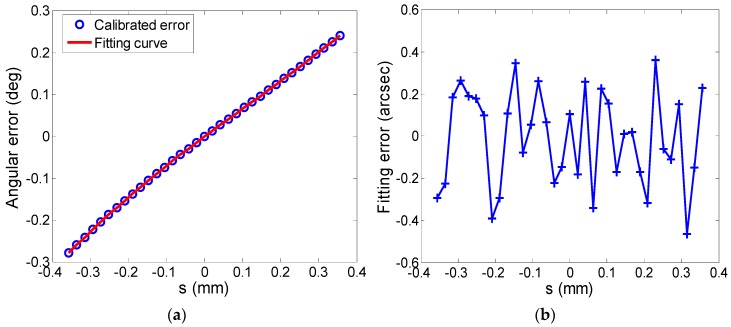
The fitting results: (**a**) Calibrated angular error and the fitting curve evaluated by the error model; (**b**) Fitting error of each calibration point.

**Figure 8 sensors-19-00367-f008:**
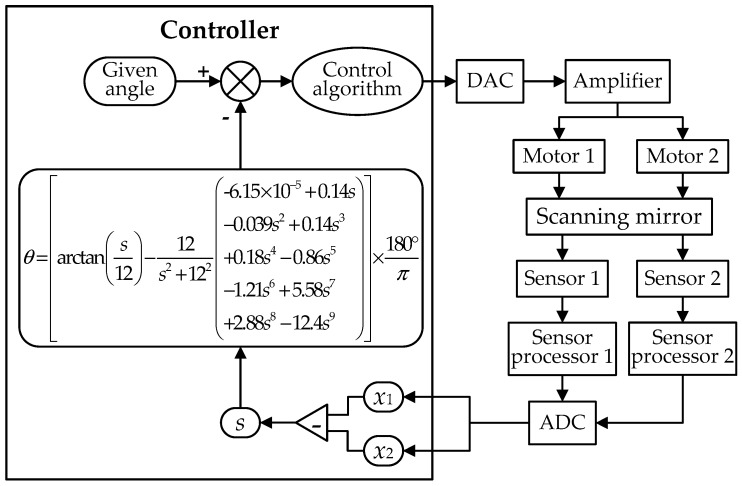
The block diagram of the control system based on the corrected angular calculation model.

**Figure 9 sensors-19-00367-f009:**
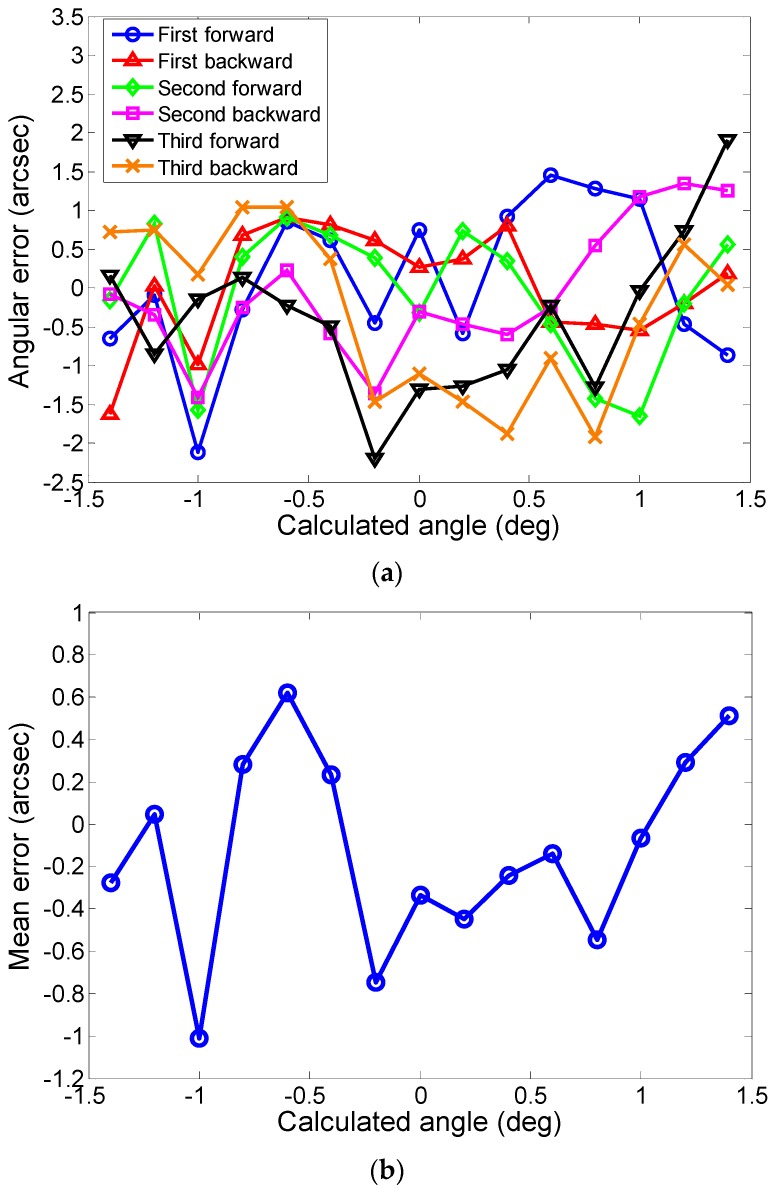
The angular errors after model correction: (**a**) Errors of the triplicate forward and backward measurements; (**b**) Mean error for each measurement point.

**Figure 10 sensors-19-00367-f010:**
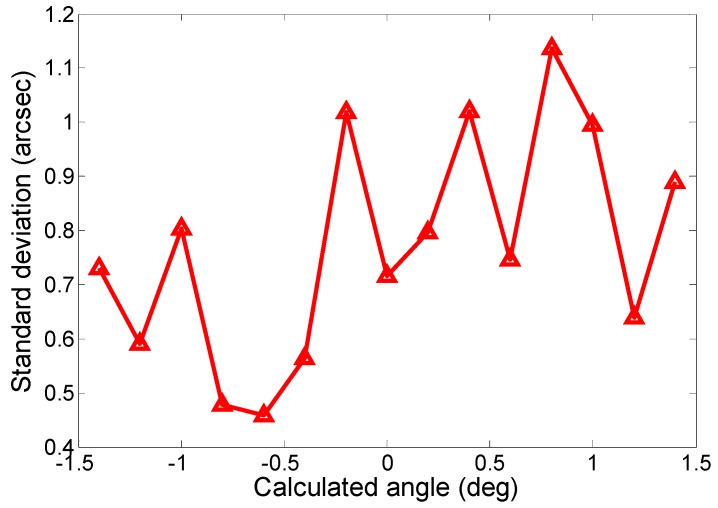
Standard deviation of each measurement point.

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
