# Peer review of "Improving Angular Accuracy of a Scanning Mirror Based on Error Modeling and Correction"

_sensors, 2019, doi:10.3390/s19020367_

Reviewer 1 Report

Review comments for Sensors-417283:

The authors propose an angular error correction method to improve the angular accuracy of the eddy current sensors in a scanning mirror. Finally, the experimental results show that the angular accuracy of the scanning mirror is dramatically improved by using the corrected calculated angles as the feedbacks of the position closed-loop control system.

In my opinion, this paper can be accepted for publication after minor revisions according to the suggested comments.

#1 In an ideal case, the axis of the rotational mirror shall be located in the center of the two eddy current sensors, please comment the variations of the center on the nonlinear errors.

#2 As you mentioned that the values of n are important for the angular error fitting…, whether the best n-values and a-parameters are also dependent on the initial maximum deflection angles for the calibration principle?

#3 In line 202, the “deference” needs to be corrected to “difference”.

Reviewer 2 Report

The paper is well organized and theresults are clearly presented. Approach is adequate an is confirmed both theoretically and by experimental studies.

At the same time the approach used in the paper looks as a straightforward application of the well known methods to specific problem. My opinion is that the manusript presents more engineering rather than scientific work.

Reviewer 3 Report

see the attached pdf file
